# Selective Parameters and Bioleaching Kinetics for Leaching Vanadium from Red Mud Using *Aspergillus niger* and *Penicillium tricolor*



**Yang Qu [1], Hui Li [1],\*, Xiaoqing Wang [1], Wenjie Tian [1], Ben Shi [1], Minjie Yao [2], Lina Cao [1] and Lingfan Yue [1]**

[1]  Department of Environmental Engineering and Chemistry, Luoyang Institute of Science and Technology, Luoyang 471023, China; quyang85@lit.edu.cn (Y.Q.); xiaoqingwang@lit.edu.cn (X.W.); twj7210@lit.edu.cn (W.T.); popping100@yeah.net (B.S.); caolina12@163.com (L.C.); ylfwyb85@163.com (L.Y.)

[2]  Key Laboratory of Environmental and Applied Microbiology, CAS, Environmental Microbiology Key Laboratory of Sichuan Province, Chengdu Institute of Biology, Chinese Academy of Sciences, Chengdu 610041, China; yaomj@cib.ac.cn

\*  Correspondence: orient.lihui@hotmail.com; Tel.: +86-379-65928271

**Abstract:** In the present study, using *Aspergillus niger* and *Penicillium tricolor*, the influence of the selected parameters, including sucrose concentration, inoculation size of spores, pulp density, and pre-culture time, on the bioleaching efficiency (biomass, organic acids production, and vanadium extraction, respectively) of red mud were studied. The bioleaching kinetics under optimal conditions were also explored. Sucrose concentration showed a positive linear effect on bioleaching efficiency below 143.44 and 141.82 g/L using *A. niger* and *P. tricolor*, respectively. However, a higher concentration was unfavorable for vanadium extraction. The inoculation size of spores showed an insignificant effect on both biomass and vanadium extraction if it exceeded the lowest coded levels ($0.5 \times 10^7$/mL). Red mud pulp density showed a negative effect on the bioleaching efficiency of *A. niger* but a positive effect on organic acids production and vanadium extraction of *P. tricolor*. A pre-culture was indispensable for *A. niger* but not for *P. tricolor* due to the fact of its isolation from the red mud examined in this study. The kinetics analysis showed that the leaching rate of vanadium followed a two-domain behavior: initially, a rapid leaching period of approximately 10–15 days and, subsequently, a slow leaching period. Considering the change of the particles' appearance as well as in the elemental composition of the bioleached red mud, it is speculated that the rate of leaching agents through the silicon minerals was the rate-limiting step of dissolution kinetics under the fungal bioleaching process.

**Keywords:** red mud; bioleaching; kinetics; fungi

## 1. Introduction

Red mud (RM) is a highly alkaline solid waste generated in the process of alumina extraction from bauxite ores [1]. The total quantity of RM stored in the world has currently reached 4 billion tons, and it is predicted to increase to 150 million tons per annum. This leads to the occupation of huge amounts of land [2–4]. Due to the alkaline, caustic, radioactive, and leaching toxicity of RM, the discharge and stockpile of RM imposes considerable environmental risks [5,6]. Therefore, it is imperative to develop a reliable way to reutilize RM.

Red mud contains various kinds of valuable metals that are worth recovering. The major metals, Al and Fe, in RM are usually recovered by reducing roasting, magnetic separation, and hydrothermal processes [7]. The minor metals, Ti, Ga, La, Y, U, and Th in RM, can be recovered by acid leaching and resin exchange [7,8]. Vanadium (V) is an important element employed in a wide range of ferrous and

non-ferrous alloys. However, studies on this topic have been rare. Previously, carbon adsorption and desorption were used to recover $V_2O_5$ from RM [9]. In recent years, a method of ion exchange with D201 resin and P507 extractant as well as a method of autoclave leaching of ferruginous hydrogarnet were reported to recover $V_2O_5$ [10,11].

Bioleaching, as an "efficient green technology" which is generally considered to be environmentally friendly and low cost, has been applied in leaching valuable metals from solid waste [12–14]. Among all types of microorganisms used in bioleaching, fungi are especially suitable for alkaline wastes leaching, because they not only survive in high alkaline–saline habitats but also secrete plentiful organic acids. *Aspergillus* and *Penicillium*, two fungal genera involved in bioleaching of solid wastes (e.g., fly ash, spent catalyst, electronic waste, and activated sludge), have been studied the most frequently [15–18].

The important factors determining fungal bioleaching efficiency are carbon source, microbial inoculation, pulp density of leaching materials, and pre-culture time of strains [19–21]. Up to now, only several studies have been performed on RM bioleaching [22–24]. However, the factors affecting the bioleaching efficiency of RM have not been studied in detail.

Generally, a higher metal leaching rate indicates a higher organic acids production of fungi, and a higher organic acids production indicates a higher fungal biomass in the bioleaching system [15,25]. How such factors influence the metal leaching rate has been previously studied [19,20]. However, how these factors simultaneously influence the three targets—metal leaching rate, organic acids production, and biomass—is still unclear. The differences in the optimum zones among these three targets also need to be studied. Response surface methodology (RSM), a collection of statistical and mathematical techniques, is a useful tool for analyzing and optimizing the independent variables in the bioleaching process.

In this study, RSM was used as the analytical model to explore how the abovementioned factors affected the bioleaching efficiency of RM. The carbon source, inoculation size of spores, pulp density of RM, and pre-culture time of strains were chosen as the selective parameters (factors). The biomass, organic acids production and extraction amount of V were defined as the optimization targets, respectively. In addition, *Aspergillus niger* and *Penicillium tricolor* were chosen as the leaching strains. Furthermore, the analysis of the leaching kinetics as well as the micromorphology of the RM particles were carried out under the optimal leaching conditions to explore the mechanism of the rate-limiting step under a fungal bioleaching process.

## 2. Materials and Methods

### 2.1. RM Samples

The RM samples generated from the Bayer process were located on the top of the RM disposal site of Chinalco in Guizhou province, China (26°41′ N, 106°35′ E). All samples were collected using sterile equipment and stored in sterile laminated stainless-steel containers. At each sampling location, six sub-samples were collected one by one from points within a horizontal distance of 10 m and used to form a composite sample. The first group of samples were used for physical–chemical analysis and bioleaching experiments. They were taken 10–30 cm below the RM residue surface and 30 m away from the dike. They were transported to the lab, dried to a constant weight in an oven at 80 °C, and crushed and powdered to 165 μm mesh. The second group of samples were used for isolating fungi. They were collected on the surface of the RM residue, 1 m away from the dike. All samples were kept at 4 °C in a refrigerator until the experiments started.

### 2.2. Fungal Strains

*Aspergillus niger* and *P. tricolor* were used as the bioleaching strains. Their nucleotide sequences were submitted to GenBank (serial number: JF909353 and JF909351, respectively). *Aspergillus niger* was provided by the Research Center for Bio-Resource & Technology, Chinese Academy of Sciences, Guiyang, China. *Penicillium tricolor* was isolated from the second group of RM samples using Horikoshi

media with 2% (*w/v*) red mud through the method of serial dilution. The Horikoshi medium comprised 10 g/L glucose, 5 g/L yeast extract, 5 g/L polypeptone, 1 g/L $K_2HPO_4$, 0.2 g/L $MgSO_4 \cdot 7H_2O$, 10 g/L $Na_2CO_3$, and 20 g/L agar and was autoclave sterilized at 121 °C for 15 min before use.

*Aspergillus niger* and *P. tricolor* were cultured on a sterilized potato dextrose agar (PDA) slant at 30 °C in a thermostatic incubator. The PDA medium comprised 200 g/L potato, 20 g/L glucose, and 20 g/L agar. The mature spores were washed off from the slant surface after 7 days culturing using a sterile solution of physiological saline (9 g/L NaCl). The number of spores was counted using a hemocytometer and standardized to approximately 0.5, 1.0, 1.5, 2.0, and $2.5 \times 10^7$/mL, respectively, with a sterilized physiological saline solution.

### 2.3. Bioleaching

Bioleaching was implemented using 250 mL Erlenmeyer flasks containing 100 mL of sterilized sucrose medium (autoclaved at 121 °C for 15 min) in an orbital shaking incubator under 120 rpm and 30 °C. A 2 mL portion of the spore suspension was added to sterilized sucrose medium. The sucrose media comprised 0.5 g/L $KNO_3$, 0.5 g/L $KH_2PO_4$, 2.0 g/L yeast extract, 2.0 g/L peptone, and sucrose with different concentration levels (list in Section 2.4).

The bioleaching experiments were deployed according to the different levels of factors (listed in Section 2.4). Samples were collected at required intervals to measure biomass, pH value, and concentration of organic acids and metal ions. The control experiments were conducted using sterilized fresh sucrose medium. The conditions of the control experiments were in accordance with case 5 and 6, respectively, in Table 1 without inoculation of the fungi.

### 2.4. Central Composite Design (CCD)

The central composite design CCD was designed to fit a reduced second-order polynomial model and to estimate the experimental error under RSM analysis. A $2^4$ factorial central CCD consisting of 16 full factorial points augmented by 8 axial points and 7 central points were created by Design-Expert 8.0, which is described in detail in a previous study [19]. A total of 31 batch bioleaching tests were performed based on the CCD.

Based on our preliminary studies about RM bioleaching [24,26], the quantitative values of five coded parameter levels (−2, −1, 0, 1, and 2) of each factor: (a) sucrose concentration was 40, 80, 120, 160, and 200 g/L, respectively; (b) inoculation size of spores was 0.5, 1.0, 1.5, 2.0, and $2.5 \times 10^7$/mL, respectively; (c) RM pulp density was 1, 4, 7, 10, and 13% (*w/v*), respectively; and (d) pre-culture time was 0, 2, 4, 6, and 8 days, respectively. The experimental conditions of pre-culture were the same as the bioleaching described in Section 2.3. But without RM. The sterilized RM was added into the culture medium when the pre-culture of the fungi was finished.

### 2.5. Kinetic Investigation

The variations of the V leaching rate, biomass, pH value, and organic acids as a function of time were measured under the optimal conditions according to the case 5 and 6 in Table 1 using *A. niger* and *P. tricolor*, respectively.

### 2.6. Analytical Methods and Data Analysis

After the required intervals, samples were taken from each flask, centrifuged at 5000 rps for 10 min and filtered through a filter membrane. The filtered liquor was analyzed for pH value, V concentration, and organic acids. The concentration of V was determined with a quadrupole inductively coupled plasma mass spectrometer (Q-ICP-MS, ELAN DRC-e, PerkinElmer, Waltham, MA, USA). The extraction percentage was calculated based on the concentration obtained from total digestion.

The concentration of organic acids was determined using high-performance liquid chromatography (HPLC, Agilent 1200, Agilent Techologies Inc., Santa Clara, CA, USA) with a variable wavelength

detector (210 nm) and Dikma-C18 column, mobile phase of 5 mM $H_2SO_4$, and flow rate of 0.5 mL/min at ambient temperature.

The biomass was determined using the same method as Aung [27] described. The residue (biomass with bioleached red mud) obtained from the filter paper was transferred to a pre-weighed evaporating dish and dried at 80 °C for 24 h, followed by ashing at 500 °C for 4 h. The dry weight of the biomass was calculated from the difference of the weight under the two temperatures.

The micromorphology of the samples was observed using scanning electron microscopy (SEM, Shimadzu-SS550, Shimadzu, Kyoto, Japan, 25 kV, 0.25 nA), and the element composition was determined by energy-dispersive X-ray spectrometry (EDS, Link ISI, Oxford Instruments, Oxford, UK). The samples were prepared by membrane filtration to remove redundant water. Then, they were washed for 1 h with 2% glutaraldehyde solution after which a series of washings with mixtures of water and ethanol were conducted for cell dehydration. Finally, the samples were coated with gold and submitted for SEM analysis.

The statistical analysis of the data, the development of the regression equation, and the drawing of the three-dimensional diagram were implemented using Design-Expert (version 8.0, Stat-Ease, Minneapolis, MN, USA). A 5% significant level was used for all statistical models. More detailed information on the analytical methods and the RM elemental composition are described in our previous studies [24,28].

## 3. Results and Discussion

### 3.1. Characteristics of RM

The pH and electrical conductivity (EC) of the RM samples were 12.9, 21.8 mS/cm, and 3.53 mmol $H^+$/g, respectively. The weight ratio of V in the studied RM samples was 0.422%, which is much higher than the average level of that (0.05–0.2%) in other Bayer and sintered RM [3,8,29]. Therefore, recovering V is especially worthy of being studied further.

### 3.2. Optimum Conditions for Maximum of Biomass, Acids Production, and V Extraction in Bioleaching

The biomass, organic acids production, and metal leaching rate generally correlated positively with each other in the bioleaching process. However, there were differences in how the factors influenced these three targets as well as in the optimum zones.

On the basis of the numerical optimization of the CCD analysis by Design-Expert, the optimum conditions for biomass, organic acids production, and V extraction are shown in Table 1. The optimum conditions were completely different from each other to achieve the three maximum targets, both for *A. niger* and *P. tricolor*. These results indicate that neither the largest biomass nor the acids production achieved the maximal metal leaching efficiency in the bioleaching process.

**Table 1.** Optimum conditions for biomass, organic acids production, and V extraction when bioleaching red mud (RM) using *Aspergillus niger* and *Penicillium tricolor* according to numerical optimization of the central composite design (CCD) analysis.

| Target | Case | Fungus | Sucrose (g/L) | Inoculation Size ($\times 10^7$/mL) | Pulp Density (%, *w/v*) | Pre-Culture Time (days) | Maximum Value of Target |
|---|---|---|---|---|---|---|---|
| Maximum of biomass | 1 | *A. niger* | 191.92 | 1.55 | 1 | 2.99 | 34.22 (g/L) |
| | 2 | *P. tricolor* | 156.36 | 1.55 | 4.88 | 5.90 | 31.48 (g/L) |
| Maximum of organic acids | 3 | *A. niger* | 174.14 | 1.81 | 5.48 | 5.50 | 109.02 (mmol/L) |
| | 4 | *P. tricolor* | 159.60 | 1.33 | 7.91 | 2.34 | 198.78 (mmol/L) |
| Maximum of V extraction | 5 | *A. niger* | 143.44 | 1.61 | 6.09 | 5.17 | 83.24 [a] (mg/L) |
| | 6 | *P. tricolor* | 141.82 | 1.29 | 8.27 | 3.80 | 119.0 [b] (mg/L) |

[a] Corresponding leaching rate of V was 32.39%; [b] Corresponding leaching rate of V was 34.10%.

Sucrose serves as an energy source of fungal growth as well as a substrate for organic acids biosynthesis in the bioleaching process. When using *A. niger* as the leaching strain, the sucrose concentration for the maximum of organic acids was lower than that for the maximum of biomass. This proved that excessive organic carbon contributed to cell reproduction but not to acid metabolism. The sucrose concentration achieving the maximal V extraction (143.44 and 141.82 g/L, respectively) was obviously lower than that achieving the maximal organic acids production (174.14 and 159.60 g/L, respectively) as well as achieving the maximal biomass (191.92 and 156.36 g/L, respectively), for both *A. niger* and *P. tricolor*. This indicates that the leached vanadium ions probably precipitated by reacting with an excess of metabolites or "sank" in hyphae through bioaccumulation and biosorption [15,30]. The best sucrose concentration of V leaching using *A. niger* and *P. tricolor* was 143.44 and 141.82 g/L, respectively. A higher sucrose concentration was unfavorable for leaching V from RM.

The RM showed a completely toxic effect on the biomass of *A. niger*, because the RM pulp density for the maximum biomass was at the negative boundary values of the CCD. But a certain amount of RM (4.88%) brought a maximum biomass of *P. tricolor*, because *P. tricolor* was isolated from RM impoundments and had a better adaptability to RM particles. The pulp density for the maximum of V extraction was obviously higher than for the maximum of biomass by both two strains. It indicated that providing enough reaction sites on the surface of RM particles was more important than reducing the cellular toxicity for the purpose of increasing the leaching efficiency of V.

Although the maximum of organic acids production of *P. tricolor* was nearly two times higher than that of *A. niger*, the maximum V extraction of the two strains were similar. This result indicates that excessive amounts of organic acids do not promote leaching efficiency significantly. The type of organic acids possibly played a more important role than the total production did. Thus, the production of organic acids could not be the only criterion for selection of leaching strains. In view of the complexity of organic acids secreted by *A. niger* and *P. tricolor*, the underlying mechanism needs to be further studied.

Previous studies showed that the biomass of *A. niger* was 21, 28, and 29 mg/L, respectively, and the total amount of organic acids was 235.6, 140, and 5.4 mmol/L, respectively, in the bioleaching system of fly ash, spent fluid catalytic cracking catalyst, and refinery hydrocracking catalyst, respectively [27,31,32]. Comparing to that, the maximum biomasses of both *A. niger* and *P. tricolor* in the RM leaching system were higher. However, the maximum of organic acids production in the RM leaching system was lower than that in the fly ash leaching system but higher than that in a refinery hydrocracking catalyst leaching system. This indicates that the different leaching materials had different impacts on fungal metabolism, even for the same fungal species.

### 3.3. Mathematical Models of Bioleaching

The statistical *p*-value, calculated from the variance analysis of CCD, can reveal the importance and correlation of factors on the targeted response value. The *p*-value of the biomass, organic acids production, and V extraction are shown in Table 2. Removing the insignificant effects ($p > 0.05$), the best possible regression models in the form of a reduced quadratic equation with respect to biomass, acids production, and V extraction are as follows (A, B, C, and D represent sucrose concentration, inoculation size of spores, RM pulp density, and pre-culture time, respectively):

$$\text{Biomass of } \textit{A. niger} = 28.24 + 1.6A - 3.72C + 3.34D - 1.21A^2 - 1.81B^2 - 1.74D^2 + 2.63CD \ (R^2 = 89.19\%) \tag{1}$$

$$\text{Biomass of } \textit{P. tricolor} = 27.31 + 3.05A - 2.47C + 1.13D - 1.13A^2 - 1.42B^2 - 1.98C^2 - 1.27D^2 + 0.83AD \ (R^2 = 95.77\%) \tag{2}$$

$$\text{Organic acids production of } \textit{A. niger} = 95.07 + 10.90A + 3.67B - 7.12C + 9.74D - 5.30A^2 - 5.09B^2 - 4.23C^2 - 7.68D^2 + 2.57AD + 3.66CD \ (R^2 = 96.40\%) \tag{3}$$

$$\text{Organic acids production of } P. \text{ tricolor} = 180.04 + 20.32A - 5.52B + 11.07C - 15.39D \\ - 11.49A^2 - 7.44B^2 - 11.04C^2 - 9.07D^2 + 6.18BC \ (R^2 = 93.59\%) \tag{4}$$

$$\text{V extraction of } A. \text{ niger} = 13.17 + 0.33A - 0.78C + 0.98D - 0.31A^2 - 1.14C2 - 0.73D^2 \\ (R^2 = 93.53\%) \tag{5}$$

$$\text{V extraction of } P. \text{ tricolor} = 14.00 + 0.67A + 0.82C - 0.60A^2 - 0.35B^2 - 1.06C^2 - 0.98D^2 \\ (R^2 = 95.36\%) \tag{6}$$

The sucrose concentration had a positive linear effect on all the targeted response values. When using a chemoheterotrophic microorganism as the leaching strain, the organic carbon source was the most important factor determining the efficiency and cost of the bioleaching process [20].

The inoculation size of the spores had an insignificant effect on the biomass and metals extraction if it exceeded the lowest coded levels ($0.5 \times 10^7$/mL) in this study. But it was significantly positive on the organic acids production of *A. niger* as well as significantly negative on that of *P. tricolor*. This means that the high inoculation size of the spores exerted a negative impact on the excretion of organic acids by *P. tricolor*. However, inoculation size was not a crucial factor for V extraction in the bioleaching system according to our results. This conclusion was consistent with the previous study with respect to the bioleaching of fly ash using *A. niger* [33].

The RM pulp density had a negative linear effect on biomass, organic acids production, and V extraction of *A. niger* but had a positive effect on organic acids production and V extraction of *P. tricolor*. This result disagreed with our assumption that RM concentration was unfavorable for all the leaching targets due to the fact of its biological toxicity [24]. A possible reason was that *P. tricolor* was isolated from the RM stockpile and, thus, a certain amount of RM exerted a stimulating effect on the fungal metabolism.

**Table 2.** Statistical *p*-value of the biomass, organic acids production, and valuable metals (V) extraction from the ANOVA of CCD using *Aspergillus niger* and *Penicillium tricolor* (A, B, C, and D represent sucrose concentration, inoculation size of spores, RM pulp density, and pre-culture time, respectively).

| Source | DF | *p*-Value of *Aspergillus Niger* | | | *p*-Value of *Penicillium Tricolor* | | |
|---|---|---|---|---|---|---|---|
| | | Biomass | Organic Acids | V Extraction | Biomass | Organic Acids | V Extraction |
| Model | 14 | ES | ES | ES | ES | ES | ES |
| Linear | 4 | ES | ES | ES | ES | ES | ES |
| A | 1 | 0.009 | ES | 0.019 | ES | ES | ES |
| B | 1 | 0.467 | 0.002 | 0.702 | 0.133 | 0.027 | 0.174 |
| C | 1 | ES | ES | ES | ES | ES | ES |
| D | 1 | ES | ES | ES | 0.001 | ES | 0.398 |
| Square | 4 | 0.003 | ES | ES | ES | ES | ES |
| A × A | 1 | 0.03 | ES | 0.019 | ES | ES | ES |
| B × B | 1 | 0.003 | ES | 0.205 | ES | 0.003 | 0.002 |
| C × C | 1 | 0.156 | ES | ES | ES | ES | ES |
| D × D | 1 | 0.003 | ES | ES | ES | ES | ES |
| Interaction | 6 | 0.062 | 0.044 | 0.467 | 0.302 | 0.231 | 0.491 |
| A × B | 1 | 0.892 | 0.091 | 0.317 | 0.748 | 0.158 | 0.635 |
| A × C | 1 | 0.645 | 0.874 | 0.755 | 0.4 | 0.846 | 0.849 |
| A × D | 1 | 0.863 | 0.049 | 0.696 | 0.021 | 0.711 | 0.704 |
| B × C | 1 | 0.964 | 0.725 | 0.438 | 0.895 | 0.041 | 0.141 |
| B × D | 1 | 0.821 | 0.972 | 0.25 | 0.955 | 0.271 | 0.141 |
| C × D | 1 | 0.001 | 0.008 | 0.131 | 0.465 | 0.451 | 0.508 |

ES: *p*-value is less than 0.0001, extremely significant.

A long pre-culture time was beneficial for the biomass, organic acids production, and V extraction when using *A. niger* as the leaching strain. This indicates that a separation between fungal cells and RM particles was indispensable before the cell growth reached the stationary phase. This result is consistent

with former studies using other leaching materials [12,34]. However, the factor of the pre-culture time for *P. tricolor* showed a negative effect on organic acids production and was insignificant for V extraction. This means that the pre-culture was not necessary for the indigenous fungal strain (*P. tricolor* was isolated from the RM stockpile) in the bioleaching process.

If the interaction among two factors is significant in the CCD model, the value of one factor changes dramatically with the variation of another one. In our experiments, when using *A. niger*, the interaction between RM pulp density and pre-culture time was extremely significant for the targeted response value of biomass as well as organic acids production. It meant there was a strong correlation between RM pulp density and pre-culture time.

Figure 1a,b display a three-dimensional response surface of V extraction as a function of pre-culture time and RM pulp density using *A. niger* and as a function of RM pulp density and sucrose concentration using *P. tricolor*, respectively (the first two important factors were chosen as the axis in light of the *p*-value). The projection point of the maximum response value is located in the plot. This illustrates that the range of the factorial value was designed appropriately. The sucrose concentration, RM pulp density, and pre-culture time had a non-linear effect on V extraction. Therefore, too high or too low of these factorial values had a negative effect on V leaching efficiency.

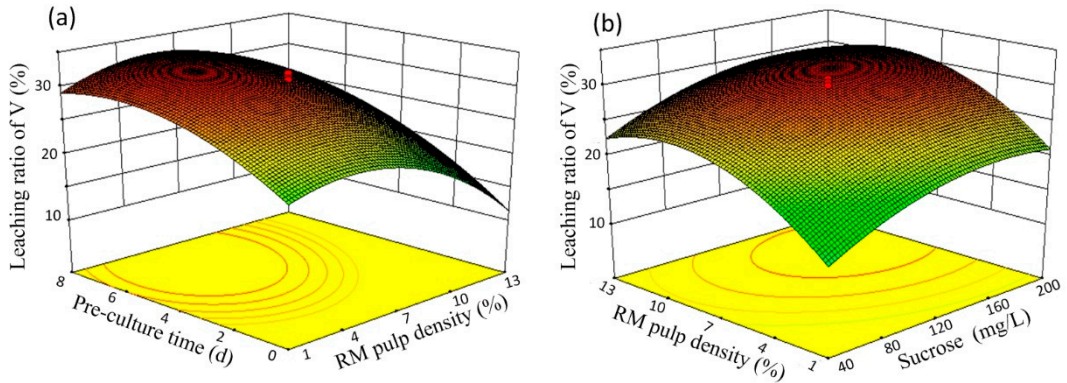

**Figure 1.** Three-dimensional response surface plot of valuable metal (V) extraction as a function of pre-culture time and RM pulp density by *A. niger* (**a**) and as a function of RM pulp density and sucrose concentration by *P. tricolor* (**b**).

### 3.4. Bioleaching Kinetics of V at Optimum Conditions

According to case 5 and 6 in Table 1, the maximum extraction amounts of V were achieved by using *A. niger* and *P. tricolor*, respectively. Therefore, the variation of the V leaching rate, biomass, pH value, and three major kinds of organic acids were determined in case 5 and 6. The sterilized fresh mediums without inoculation of fungi were used as the control experiments.

### 3.4.1. Variation of V Leaching Rate

The variation of the V leaching rate is shown in Figure 2. In the bioleaching system, the leaching process of V began after the pre-culture finished and the RM was added into the culture. The leaching rate suddenly increased to 8–13% in two days, which was due to the rapidly acidolysis of organic acids that were already secreted by fungi in the pre-culture period. The entire bioleaching kinetics, by and large, followed a two-domain behavior: initially, a rapid leaching period (10–15 days) and, subsequently, a slow leaching period (20 days). A previous study reported that the rapid period was only 3–7 days for leaching a spent refinery catalyst by *A. niger* [19]. It was probably because the biological toxicity of the spent refinery catalyst was much lower than that of the RM. The transition of the rapid period to the slow period was possibly caused by the depletion of active organic acids, accumulation of toxic metabolic substance, losses of available reacting sites, and formation of a non-porous product layer on RM particles. Though the biomass and organic acids production obviously decreased at the end of

leaching process, the V leaching rate increased slightly. It was probably because the V ions adsorbed or accumulated on fungal cells were released into the solution after cellular hydrolysis.

Although the total leaching rate of V was slightly lower compared to that under a continuous leaching mode at 2% RM pulp density (37%) in our former study [24], the extraction amount of V was higher.

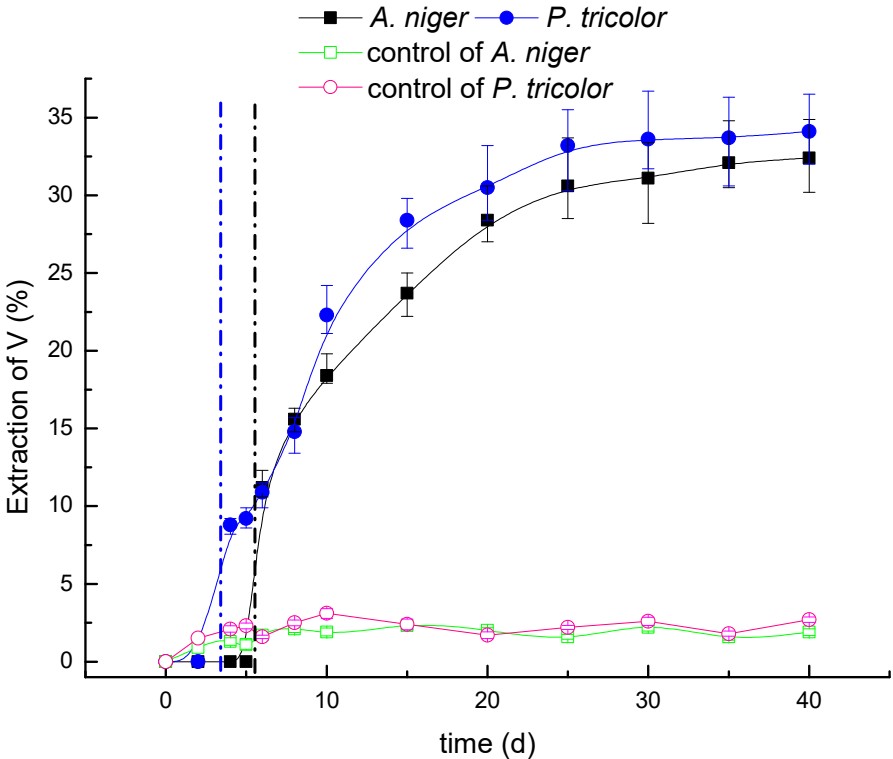

**Figure 2.** Variation of the leaching rate of V as a function of time using *A. niger* and *P. tricolor* under optimal leaching conditions (dashed and dotted lines represent the time when the pre-culture finished and RM was added into leaching system using *A. niger* (black) and *P. tricolor* (blue)).

### 3.4.2. Variation of Biomass and pH Value

The change of biomass and pH value as a function of time is shown in Figure 3. The pH of fresh medium (control experiments) increased to 12.5 and 12.8, respectively, due to the release of alkaline ions from RM. The biomasses of control experiments were zero because there was no inoculation of the fungi (data not shown).

In the bioleaching system, the variation of the fungal biomass underwent 4 phases: first growing phase (pre-culture period); lag phase; second growing phase; and decline phase. The biomass increased rapidly in the first growing phase. After the RM was added, *A. niger* and *P. tricolor* underwent a lag phase of 10 and 5 days, respectively, to acclimate. Then, the biomass reached the maximum in the second fast growing period. The biomass had an obvious decrease at the end of the leaching process due to the cellular hydrolysis.

The pH value had a highly negative correlation with biomass which indicates that the metabolic activity of the fungal strains played a key role in controlling the pH value of the leaching solution. The pH value increased within the first 4–5 days of the lag phase after RM was added, which means that the production of active organic acids from fungi was slower than the dissolution of alkaline ions ($OH^-$, $CO_3^{2-}$, and $HCO_3^-$) from RM minerals (tri-calcium aluminate, sodium–aluminum–silicate, calcite, etc.). However, due to the gradual depletion of the alkaline minerals in the RM, along with an increase in the organic acids production via fungal acclimation, the pH value of the leaching system decreased again.

The maximum biomass of *P. tricolor* (28.6 mg/L) was higher than that of *A. niger* (25.4 mg/L) under the optimal V leaching conditions (case 5 and 6, respectively, in Table 1). However, under the optimal conditions for biomass (case 1 and 2, respectively, in Table 1), the maximum biomass of *P. tricolor* (31.48 mg/L) was lower than that of *A. niger* (34.22 mg/L). This result proved that the maximum biomass should not be the most important parameter designed for fungal bioleaching, though it had a positive correlation with metal leaching efficiency as well as organic acids production.

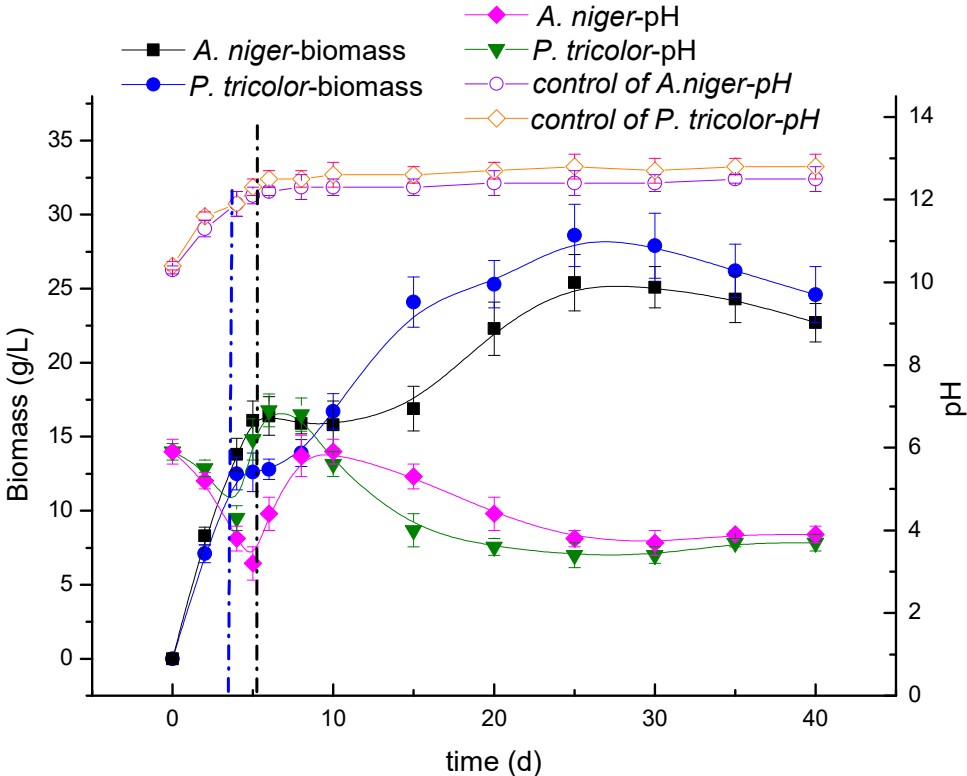

**Figure 3.** Variation of the biomass and pH values as a function of time using *A. niger* and *P. tricolor* under optimum V leaching conditions (dashed and dot lines represent the time when the pre-culture finished and the RM was added into the leaching system using *A. niger* (black) and *P. tricolor* (blue), respectively).

### 3.4.3. Variation of Organic Acids Production

The HPLC chromatogram of *P. tricolor* after bioleaching for 30 days is shown in Figure 4. A large number of peaks appeared on the chromatogram which indicates that various kinds of metabolites were secreted by fungi. The two highest peaks on the chromatogram represented oxalic and citric acids. The peaks of gluconic acids were inconspicuous due to the negligible production.

The variation of organic acids production as a function of time during the bioleaching process is shown in Figure 5. In the pre-culture period, *A. niger* mainly secreted citric acid, and *P. tricolor* mainly secreted gluconic acid. After RM was added into the culture, the production of oxalic acid accelerated rapidly until it reached the maximum level for both *A. niger* and *P. tricolor*. Then, the production of oxalic acid gradually decreased in the decline phase but was still the largest one in the leaching system. This result proved that the RM had a strong stimulating effect on oxalic excretion of *A. niger* and *P. tricolor*. A previous study reported that, for leaching power plant residual ash, oxalic acid was the main organic acid secreted by *A. niger* if the pH value exceeded 4.0 [21] which is consistent with our study.

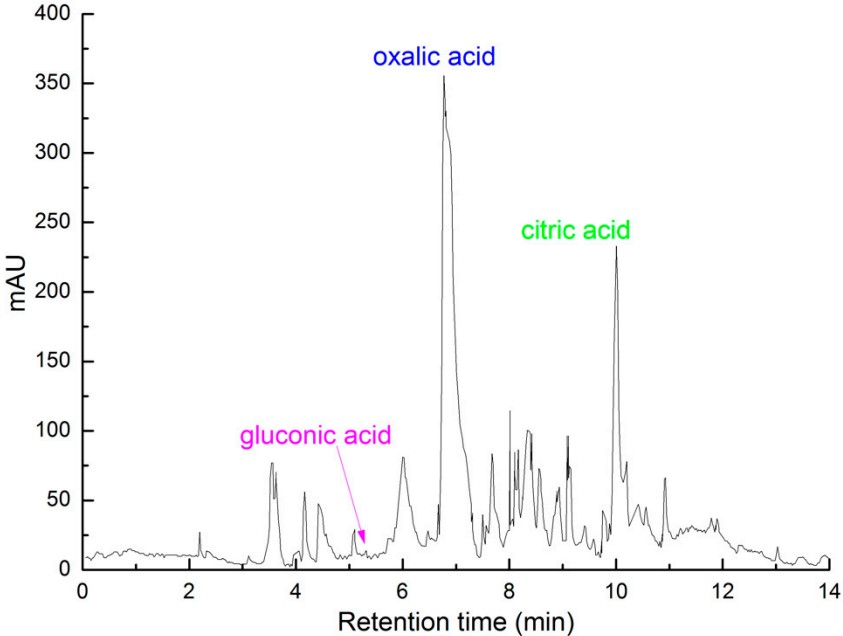

**Figure 4.** Chromatogram of HPLC using *P. tricolor* on the 30th day under optimal bioleaching conditions.

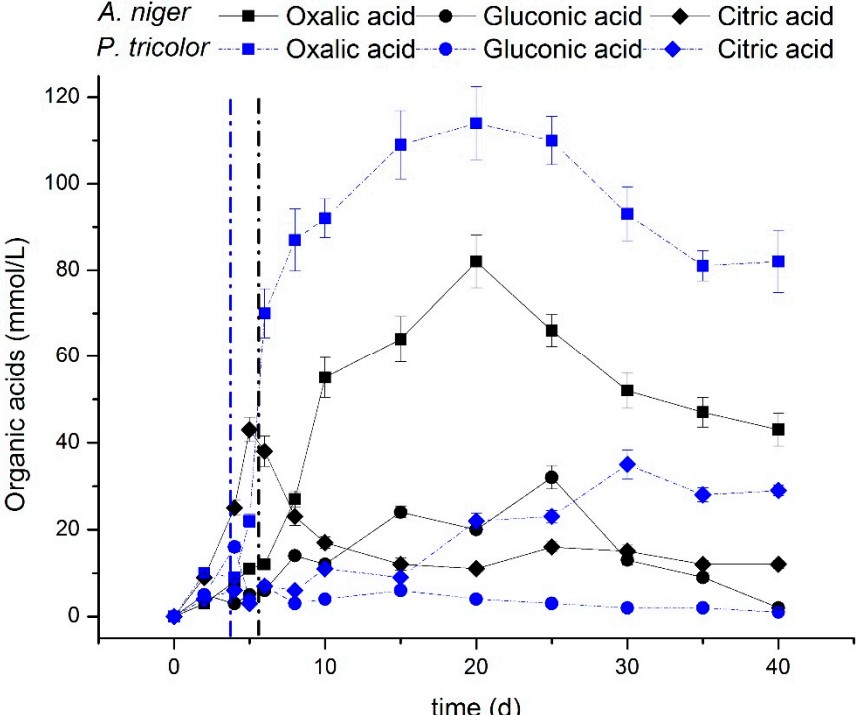

**Figure 5.** Variation of the concentration of oxalic, gluconic, and citric acids as a function of time using *A. niger* and *P. tricolor* under optimum V leaching conditions (dashed and dotted lines represent the time when the pre-culture finished and the RM was added into the leaching system using *A. niger* (black) and *P. tricolor* (blue), respectively).

At the end of the leaching process, the concentration of gluconic acid decreased to approximately zero, but there was still a small quantity of citric acid in the medium for both *A. niger* and *P. tricolor*. The secretion of gluconic acid was controlled by an extracellular glucose oxidase enzyme formed in the initial phase of fermentation. It was inhibited with a pH value lower than 3.5 [35]. The secretion

of citric acid was accompanied by a decrease in oxalic acid production due to the consumption in the citric acid production cycle known as the Krebs cycle which usually occurs after the exponential growth phase at a lower external pH value [21].

In conclusion, following the decrease in the pH value in the leaching system, the production order of the organic acids by the two fungi was oxalic acid, then gluconic acid, and lastly citric acid.

## 3.5. Micromorphology Analysis of Fungi and RM Particles

The SEM photomicrographs of fungi and RM particles before and after leaching are displayed in Figure 6. The mycelium pellets of *P. tricolor* were homogeneously oval or round, and the diameter was approximately 500–700 μm (Figure 6a). The width of the mycelia was between 2 μm and 10 μm. During the bioleaching process, most of the mycelia were covered by RM particles (Figure 6b).

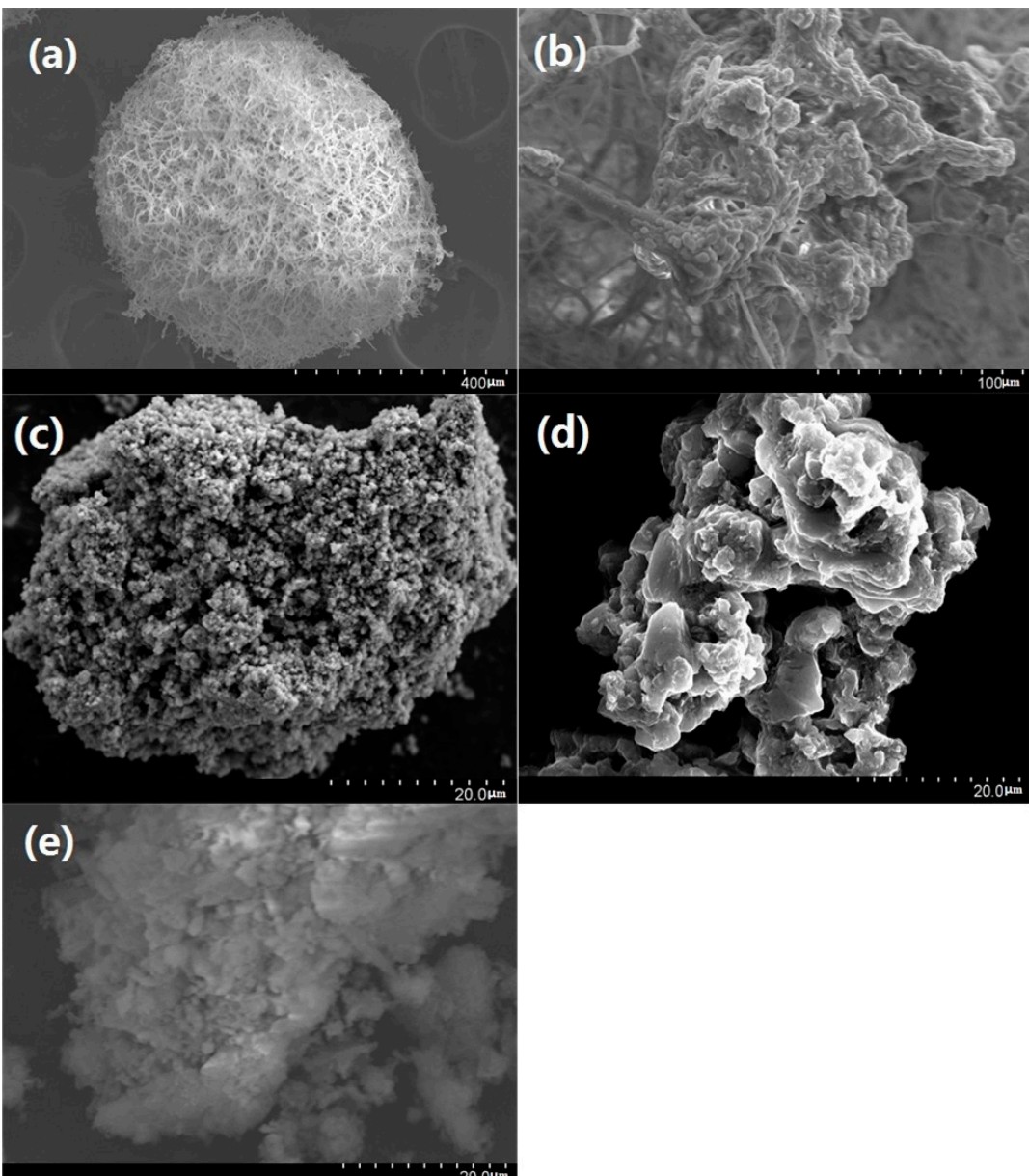

**Figure 6.** SEM images: (**a**) mycelium pellet of *P. tricolor*; (**b**) RM particles adhering to mycelium of *P. tricolor* during the bioleaching process; (**c**) raw RM particles; (**d**) RM particles after 40 h bioleaching using *P. tricolor* according to the condition of case 6 in Table 1 (red arrows show the erosion traces); (**e**) RM particles after 40 h leaching using fresh medium without fungi (control experiment).

The raw RM were comprised of various particles and aggregates with greatly differing sizes ranging from 0.05 μm to over 50 μm [14]. The RM particles with a diameter of approximately 50 μm were chosen as the representatives of the SEM analysis. The raw RM particles were completely covered by fine particles that were smaller than 1.0 μm (Figure 6c). After bioleaching of *P. tricolor*, a large quantity of fine particles on the surface of RM disappeared, leaving a relatively smooth and layered structure (Figure 6d). The erosion traces imprinted by fungi were clearly shown on the bioleached RM particles. The holes (marked by red arrows in Figure 6d) "dug" by fungi were probably due to the chemical erosion by metabolites (such as organic acids and $CO_2$) as well as physical destruction by mycelia growth. The micromorphology of the RM that was leached by *A. niger* was almost the same as that by *P. tricolor* (data not shown). The appearance of the RM that was leached by fresh medium without inoculation of fungi did not obviously change compared to raw RM (Figure 6e).

The EDS results showed that, after bioleaching, the weight percentage of V, Fe, Ca, Na, K, and Al decreased (Table 3). This indicates that these metal elements were leached into the solution efficiently via fungal activity. However, the weight percentage of Si increased from 5.6% to 12.0% after bioleaching, which indicates that Si was very difficult to leach and remained in the bioleached RM.

Previous studies [36,37] have confirmed that fine RM particles are indicative of an iron oxide phase (e.g., hematite), while large and smooth particles are silicon compounds (e.g., sodalite and cancrinite). In view of the change in the micromorphology (i.e., disappearance of fine particles and emergence of a smooth layer on the surface of bioleached RM) as well as the change in the element composition of RM, it was concluded that iron oxides easily erode, while silicon compounds are difficult to erode or destroy by fungi activity.

**Table 3.** EDS analysis of raw and bioleached RM.

| Materials [a] | V | Fe | Ca | Na | K | Al | Si |
|---|---|---|---|---|---|---|---|
| Raw RM (wt %) [b] | 0.4 ± 0.1 | 7.2 ± 0.6 | 15.1 ± 1.6 | 2.6 ± 0.3 | 1.0 ± 0.2 | 3.3 ± 0.3 | 5.6 ± 0.7 |
| Bioleached RM (wt %) [c] | 0.2 ± 0.1 | 4.6 ± 0.5 | 9.6 ± 1.1 | 0.7 ± 0.2 | 0.5 ± 0.1 | 1.6 ± 0.1 | 12.0 ± 1.0 |

[a] The mean weight percentage of elements in RM ($n$ = 3); [b] The first group of RM samples described in Section 2.1. without bioleaching; [c] The RM particles were sampled from the optimized bioleaching system according to case 6 in Table 1 after 40 h leaching using *P. tricolor*.

## 4. Conclusions

The optimum conditions of the four selected parameters (sucrose concentration, inoculation size of spores, RM pulp density, and pre-culture time) were completely different in order to achieve the three maximum targets (i.e., biomass, acids production, and V extraction) for both *A. niger* and *P. tricolor*. This indicates that the increase in the metal leaching efficiency might not only depend on optimizing the acids production and biomass of fungi. Sucrose concentration was one of the most important parameters controlling the operating cost of bioleaching. The sucrose concentration of maximal V extraction was obviously lower than that of maximal organic acids production and biomass which was conducive to the decrease in the bioleaching cost. The RM pulp density and pre-culture experiments showed that *P. tricolor* tolerated higher RM pulp densities and needed shorter pre-culture times compared to *A. niger*. This was because *P. tricolor* was isolated from the RM samples and probably had better survivability in terms of high saline–alkaline stress.

The bioleaching kinetics study showed that the leaching rate of V followed a two-domain behavior: initially, a rapid leaching period and, subsequently, a slow leaching period. The bioleaching process can be controlled in the rapid leaching period and, thus, the duration could be shortened from 40 days to 15 days in order to reduce the time–cost from an industrial standpoint. After bioleaching, the fine particles (i.e., iron oxides) disappeared, while a smooth and layered structure (silicon compounds) emerged on the surface of bioleached RM particles. Considering that the weight percentages of V, Fe, Ca, Na, and K decreased and that of Si increased in the bioleached RM, it was concluded that iron oxides are easily eroded while silicon compounds are difficult to erode or destroy by fungi activity.

Therefore, it was speculated that the rate of leaching agents (mainly organic acids) through the silicon minerals was the rate-limiting step of dissolution kinetics under fungal bioleaching processes.

For the purpose of increasing the total metal leaching rate, the efficient erosion and destruction of silicon compounds in RM particles should be investigated in the future, e.g., combine chemical leaching (inorganic acids) or bio-desilication (*Bacillus mucilaginosus*) into the fungal bioleaching process.

**Author Contributions:** Experiments, Y.Q., H.L., L.C., L.Y.; Writing—Original Draft Preparation, Y.Q.; Writing—Review & Editing, H.L., X.W., W.T., B.S. and M.Y.

**Funding:** This work was jointly supported by the National Natural Science Foundation of China (41701306, 51804155, 41701358), Breakthrough of Science and Technology of Henan Province (182102311035), Key Scientific Research Projects of University in Henan (17B610007), and Innovation and Entrepreneurship Competition of College Students (201911070006).

**Conflicts of Interest:** The authors declare no conflict of interest.

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
