# Peer review of "Selective Parameters and Bioleaching Kinetics for Leaching Vanadium from Red Mud Using Aspergillus niger and Penicillium tricolor"

_minerals, doi:10.3390/min9110697_

Round 1
Reviewer 1 Report
Manuscript minerals-621306
The manuscript by Qu et al. presents some interesting information on the bioleaching of vanadium from red mud using fungi. In my opinion the manuscript covers too many aspects of these process, maybe the modeling on the shrinking of the particles could be deleted and perhaps presented in another manuscript.
The use of the term “ratio” is unclear to me, I don’t know if they mean “rate” or what. Another important point is related with the units of the sugar concentration, mg/l or g/l. If it is g/l then the biomass obtained should be revised because the media does not contain enough nitrogen source to produce near 200 g cell mass/l.
English should be revised.
Author Response
Point 1: The manuscript by Qu et al. presents some interesting information on the bioleaching of vanadium from red mud using fungi. In my opinion the manuscript covers too many aspects of these process, maybe the modeling on the shrinking of the particles could be deleted and perhaps presented in another manuscript.
Response 1: The study of shrinking core model can explain the mechanism of rate controlling step of V leaching. We considered that it was useful to explore the mechanism of bioleaching kinetics. However, this content can be deleted in our manuscript if you believe it is not necessary for our manuscript. Thank you!
Point 2: The use of the term “ratio” is unclear to me, I don’t know if they mean “rate” or what.
Response 2: Yes, you are right! It means “rate” or “percent”. We have revised “leaching ratio” to “leaching rate” in our manuscript. Thank you very much!
Point 3: Another important point is related with the units of the sugar concentration, mg/l or g/l. If it is g/l then the biomass obtained should be revised because the media does not contain enough nitrogen source to produce near 200 g cell mass/l.
Response 3:
The units of the sugar concentration was g/L. We have revised in section 2.4.
The maximum biomass of A. niger and P. tricolor in optimal leaching conditions (Table 1) were 34.22 and 31.48 g/L, respectively. They were much smaller than 200 g/L. Thank you!
Point 4: English should be revised.
Response 4: The English were thoroughly and carefully revised by several experts. Thank you!
Reviewer 2 Report
Minor comments:
2 rephrase „influencing factors“; e.g. to „selective parameters“
14 Aspergillus niger and Penicillium tricolor in italic
17 what are the „optimal conditions“? never stated throughout the manuscript
18 showed a positive; excessive sucrose addition
19 fungi in italic
20 “indecisive factor”? meaning?
21 a negative; a positive
23 …. but not for P. tricolor due to its isolation from the red mud examined in this study
24 two domain behavior à high & slow leaching rate?
27-33 redundant
48&49 … however, studies on this topic have been rare
81&85 don’t understand statement, easier phrasing?
95-97 in which city/state/province is disposal site located? For isolation of indigenous fungus, where was sample taken from? Surface, bottom, beneath surface… if bottom, than please state a depth from the surface.
99-105 more details on culturing; numerical composition of potato agar? How were fungi cultivated – temperature, oxygen, pH? How were fungi counted/enumerated?
107-112 which sucrose levels were applied? Or say “listed in 2.4”
131 bioleached residues = fungi biomass after bioleaching?
140 other data on RM elemental composition?
166-169 link the statement to table, makes it more understandable were claim comes from
172 should be “a toxic”
204 spores don’t count as biomass? What is fungal biomass then?
245 what are case 5 and 6 in table one? Specify in table one with number/letter etc
285-287 what are the optimal conditions for V extraction/biomass??
381 weight% from eds? Table?
396 in which state was raw red mud examined? Dried first, native state, freeze-dried etc.
398 conclusion is basically a recap of results, no interpretation, needs to be developed
413-415 can Si layer be dissolved, and fungal activity be increased again? Can formation of inhibiting layer be avoided? How can strains be optimized, what are next steps, etc
Table 1. Which kind of “Value”, which units of it? please decipher in the table
Major problems:
373-380 All this paragraph is not supported by the presented figure. Please provide SEM images of the same magnification: SEM of raw RM has to be done at the same magnification with SEM of bioleached RM. Furthermore, the authors need to present more data and images on high resolution SEM to suggest any crystallinity of any structures. Without XRD measurements one cannot talk about crystal structures. It might be possible to talk about “erosion traces” in case with high resolution, good quality and resolution compatibly of SEM images.
381-388 EDS spot measurements are not acceptable in this case, this is not a valid approach applicable towards bioleaching. First of all, presenting results from a single spot measurement does not cover the whole picture with your RM, which is quite heterogeneous substance. The authors should provide the results of the bulk analysis of solid raw and solid bioprocessed RM. This will be a proper measure of bioleaching efficiency derived from solid materials.
413-415 Consequently, conclusion and abstract too have to be reconsidered based on previous major comments.
More information is needed on how the mean diameter of RM particles was measured, perhaps any SEM images of these particles before and after the fungi?
Please provide all the conditions of your pre-culture experiments: which medium, conditions, with or without RM?? This is not clear in the manuscript.
I did not find any abiotic control experiments in the manuscript. Which negative (abiotic) control did the authors use to show the influence of physico-chemical parameters used in their cultivation (e.g., chemicals in cultivation medium, pH, etc.)?? Abiotic controls need to be always represented for all kind of experiments: SEM, IPC-MS, particles size distribution, etc. We usually use the culture medium kept under the cultivation temperature and pH with raw metal-bearing materials, but without microorganisms. What was used in your study as abiotic control? And please provide the results of it. 

Please provide HPLC 
profiles/peaks of detected oxalic, gluconic, and citric acids.
A lot on the grammar and vocab need to be improved.
Author Response
Point 1: 2 rephrase „influencing factors“; e.g. to „selective parameters“
Response 1: We have already rephrased “influencing factors” to “selective parameters” in our manuscript. Thank you!
Point 2:14 Aspergillus niger and Penicillium tricolor in italic
Response 2: We have corrected this error. Thank you very much!
Point 3:17 what are the optimal conditions“? never stated throughout the manuscript
Response 3: The optimal conditions of V bioleaching by using A. niger and P. tricolor were shown in case 5 and 6 respectively in Table 1. And they were also stated in the first line in section 3.4. Thanks.
Point 4: 18 showed a positive; excessive sucrose addition
Response 4: The description was not explicit in our manuscript. Therefore we have revised to “Sucrose concentration showed a positive linear effect on bioleaching efficiency if below 143.44 and 141.82 g/L by using A. niger and P. tricolor respectively. However higher was unfavorable for vanadium extraction.” Thank you!
Point 5: 19 fungi in italic
Response 5: We have corrected. Thank you!
Point 6: 20 “indecisive factor”? meaning?
Response 6: We have revised this sentence and deleted the ‘indecisive factor’. Thank you!
Point 7:21 a negative; a positive
Response 7: We have corrected in our manuscript. Thanks a lot!
Point 8: 23 …. but not for P. tricolor due to its isolation from the red mud examined in this study
Response 8: We have revised this sentence. Thank you very much!
Point 9:24 two domain behavior à high & slow leaching rate?
Response 9: We have revised it in our manuscript as: “The kinetics analysis showed that leaching rate of vanadium followed a two-domain behavior (an initially rapid leaching period about 10-15 days and a subsequently slow leaching period).” Thank you!
Point 10: 27-33 redundant
Response 10: This is our fault. We have revised it. Thanks a lot!
Point 11: 48&49 … however, studies on this topic have been rare
Response 11: We have corrected this sentence. Thank you very much!
Point 12: 81&85 don’t understand statement, easier phrasing?
Response 12: We have revised this paragraph to make it clear. “The leaching agent concentration in fungal leaching system is not constant. Thus the simulation of leaching kinetics by traditional models will be inaccurate. However, the modified equations of shrinking core model have been specially developed to describe the leaching kinetics in the case that the protons change dramatically [27]”. Thank you!
Point 13:9 5-97 in which city/state/province is disposal site located? For isolation of indigenous fungus, where was sample taken from? Surface, bottom, beneath surface… if bottom, than please state a depth from the surface.
Response 13: We have described the methods and RM samples location in detail as follows.
“The RM samples generated from Bayer process were located on the top of RM disposal site of Chinalco in Guizhou province, China (26º41’N, 106º35’E). All the samples were collected using sterile equipment and stored in sterile laminated stainless steel containers. At each sampling location, six sub-samples were collected one by one from points within a horizontal distance of 10 m and used to form a composite sample. The first kind of samples were used for physical-chemical analysis and bioleaching experiments. They were taken 10–30 cm below the RM residue surface and 30 m away from the dike. They were transported to the lab, dried to constant weight in an oven at 80°C, crushed and powdered to 165 μm mesh. The second kind of samples were used for isolating fungi. They were collected on the surface of RM residue and 1 m away from the dike. All the samples were kept at 4°C in refrigerator until the experiments started.”
Thank you!
Point 14: 99-105 more details on culturing; numerical composition of potato agar? How were fungi cultivated – temperature, oxygen, pH? How were fungi counted/enumerated?
Response 14: We have revised this section as follows. All the methods and materials were described in detail. Thank you!
"A. niger and P. tricolor were used as the bioleaching strains. Their nucleotide sequences were submitted to GenBank (serial number: JF909353 and JF909351, respectively). A. niger was provided by the Research Center for Bio-Resource & Technology, Chinese Academy of Sciences. P. tricolor was isolated from the second kind of RM samples using Horikoshi media with 2% (w/v) red mud through the method of serial dilution. The Horikoshi medium comprised 10 g/L glucose, 5 g/L yeast extract, 5 g/L polypeptone, 1 g/L K2HPO4, 0.2 g/L Mg2SO4∙7H2O, 10 g/L Na2CO3, and 20 g/L agar, and was autoclave sterilized at 121°C for 15 min before use.
A. niger and P. tricolor were cultured on sterilized potato dextrose agar (PDA) slant at 30°C in thermostatic incubator. The PDA medium comprised 200 g/L potato, 20 g/L glucose, and 20 g/L agar. The mature spores were washed off from the slant surface after 7 days culturing using a sterile solution of physiological saline (9 g/L NaCl). The number of spores was counted using a haemocytometer, and standardized to approximately 0.5, 1.0, 1.5, 2.0, and 2.5×107/mL respectively with a sterilized physiological saline solution.”
Point 15:107-112 which sucrose levels were applied? Or say “listed in 2.4”
Response 15: We have revised this sentence as “The sucrose media used in the tests comprised 0.5 g/L KNO3, 0.5 g/L KH2PO4, 2.0 g/L yeast extract, 2.0 g/L peptone, and sucrose with different concentration levels (list in section 2.4).” We have revised all this paragraph to make it clear as follows.
“Bioleaching were implemented by using 250 mL Erlenmeyer flasks containing 100 mL of sterilized sucrose medium (autoclaved at 121℃ for 15 min) in an orbital shaking incubator under 120 rpm and 30℃. A 2 mL portion of the spore suspension was added to sterilized sucrose medium. The sucrose media comprised 0.5 g/L KNO3, 0.5 g/L KH2PO4, 2.0 g/L yeast extract, 2.0 g/L peptone, and sucrose with different concentration levels (list in section 2.4).
The bioleaching experiments were deployed according to the different levels of factors (listed in section 2.4). Samples were collected at required intervals to measure biomass, pH value, and concentration of organic acids and metal ions. The control experiments were conducted using sterilized fresh sucrose medium. The conditions of control experiments were in accordance with the case 5 and 6 respectively in Table 1 without inoculation of fungi. “
Thank you!
Point 16:131 bioleached residues = fungi biomass after bioleaching?
Response 16: Bioleached residues means the fungal biomass adhered with RM particles after bioleaching. Thanks.
Point 17:140 other data on RM elemental composition?
Response 17: Yes, more data on RM elemental composition and detailed information of analytical methods in our previous study. We have revised this sentence to “The more detailed analytical methods and RM elemental composition were described in our previous study [24, 29]”. Thank you!
Point 18:166-169 link the statement to table, makes it more understandable were claim comes from.
Response 18: We have used the data (shown in Table 1) in the sentence to make our statement more clearly. Now the statement is “The sucrose concentration achieving the maximal V extraction (143.44 and 141.82 g/L, respectively) was obviously lower than that achieving the maximal organic acids production (174.14 and 159.60 g/L, respectively) as well as achieving the maximal biomass (191.92 and 156.36 g/L, respectively), both for A. niger and P. tricolor respectively.” Thank you very much!
Point 19:172 should be “a toxic”
Response 19: Sorry, this is our fault! We have corrected it. Thank you!
Point 20:204 spores don’t count as biomass? What is fungal biomass then?
Response 20: The term “spores concentration” is not precise in here. It means the inoculation size (or inoculation concentration) of fungal spores. The biomass of inoculative spores was negligible comparing to the biomass in a running bioleaching system. We have corrected “spores concentration” to “inoculation size of spores” in our entire manuscript. Thank you very much!
Point 21: 245 what are case 5 and 6 in table one? Specify in table one with number/letter etc
Response 21: We have specified in Table 1 with number. Thank you!
Point 22: 285-287 what are the optimal conditions for V extraction/biomass??
Response 22: Optimal conditions for V were shown in case 5 and 6 by using A. niger and P. tricolor respectively. Optimal conditions for biomass were shown in case 1 and 2 by using A. niger and P. tricolor respectively. We have revised this paragraph to make it easy to understand.
“The maximum biomass of P. tricolor (28.6 mg/L) was higher than that of A. niger (25.4 mg/L) under the optimal V leaching conditions (case 5 and 6 respectively in Table 1). However, under the optimal conditions for biomass (case 1 and 2 respectively in Table 1), the maximum biomass of P. tricolor (31.48 mg/L) was lower than that of A. niger (34.22 mg/L). This result proved that the maximum biomass should not be the most important parameter designed for fungal bioleaching, though it had a positive correlation with metal leaching efficiency as well as organic acids production.”
Thank you!
Point 23: 381 weight% from eds? Table?
Response 23: We have deleted the EDS results according to your comments (Point 28). Thank you!
Point 24: 396 in which state was raw red mud examined? Dried first, native state, freeze-dried etc.
Response 24: Dried to constant weight in an oven at 80℃, crushed and powdered to 165 μm mesh, and stored at 4°C in refrigerator. We have added the explanation about raw red mud in the footnote in Table 3. And we also have given the detailed information in section 2.6. Thank you very much!
Point 25:398 conclusion is basically a recap of results, no interpretation, needs to be developed
Response 25: We have totally revised the conclusion. Thank you for your suggestion!
Point 26:413-415 can Si layer be dissolved, and fungal activity be increased again? Can formation of inhibiting layer be avoided? How can strains be optimized, what are next steps, etc
Response 26: We have revised the content about future work in our manuscript in conclusion. Thank you!
Point 27:Table 1. Which kind of “Value”, which units of it? please decipher in the table
Response 27: “Value” means the “Maximum value of target”. We have already revised it in Table 1. The units were shown in the first column in this table.
Major problems:
Point 28: 373-380 All this paragraph is not supported by the presented figure. Please provide SEM images of the same magnification: SEM of raw RM has to be done at the same magnification with SEM of bioleached RM. Furthermore, the authors need to present more data and images on high resolution SEM to suggest any crystallinity of any structures. Without XRD measurements one cannot talk about crystal structures. It might be possible to talk about “erosion traces” in case with high resolution, good quality and resolution compatibly of SEM images.
Response 28: Your suggestion is very helpful for us. We have replaced our SEM images. Now the magnification of raw RM is the same as that of bioleached RM. We have deleted the description about crystal structures in our manuscript. We will do the XRD experiments if you think this is indispensable for our manusript. We have also added the SEM graphs of mycelia pellets with and without RM. Thanks a lot!
Point 29: 381-388 EDS spot measurements are not acceptable in this case, this is not a valid approach applicable towards bioleaching. First of all, presenting results from a single spot measurement does not cover the whole picture with your RM, which is quite heterogeneous substance. The authors should provide the results of the bulk analysis of solid raw and solid bioprocessed RM. This will be a proper measure of bioleaching efficiency derived from solid materials.
Response 29: You are right! We appreciate your suggestion about the EDS analysis. We have deleted the results of EDS analysis in our manuscript. But only 10 days we have to revise our manuscript. We need more time to do the bulk analysis of RM if you consider this is very important for this paper. Thank you very much!
Point 30: 413-415 Consequently, conclusion and abstract too have to be reconsidered based on previous major comments.
Response 29: The abstract and conclusion have been thoroughly revised. Thank you!
Point 31: More information is needed on how the mean diameter of RM particles was measured, perhaps any SEM images of these particles before and after the fungi?
Response 31: The experimental details about measuring diameter were added into our manuscript in section 2.6 as follows:“The particle sizes of raw and bioleached RM were measured by laser granulometer (Microtrac S3500+SDC). The raw RM were the first kind of RM samples after dried and crushed described in section 2.1. The bioleached RM were sampled from the optimized bioleaching system according to case 5 and 6 respectively in Table 1 after 40 h leaching using A. niger and P. tricolor respectively. The residue under the bottom of flask were collected by filter membrane (0.2 μm, Whatman). The bioleached RM particles were collected by washing the residue with deionized water, centrifuged at 3000 r/min, and dried to constant weight at 80°C. The RM samples (5 g) were mixing with 50 mL dispersing agent (deionized water), and injected into laser granulometer using a 10 mL dropper. ”
The RM particles with the length of approximate 50 μm were chosen as the representative of SEM analysis, because they were easier to be observed comparing to very fine particles.
Point 32: Please provide all the conditions of your pre-culture experiments: which medium, conditions, with or without RM?? This is not clear in the manuscript.
Response 32: The experimental conditions of pre-culture were the same as the bioleaching conditions described in section 2.3 but without RM. We have added the detailed information about pre-culture experiments in section 2.4. We also revised the details of bioleaching experiments in section 2.3 to make them clear.
Point 33: I did not find any abiotic control experiments in the manuscript. Which negative (abiotic) control did the authors use to show the influence of physico-chemical parameters used in their cultivation (e.g., chemicals in cultivation medium, pH, etc.)?? Abiotic controls need to be always represented for all kind of experiments: SEM, IPC-MS, particles size distribution, etc. We usually use the culture medium kept under the cultivation temperature and pH with raw metal-bearing materials, but without microorganisms. What was used in your study as abiotic control? And please provide the results of it.
Response 33: Your suggestion is right! We have used the fresh sterilized sucrose medium as the control experiments. But they were not shown in our previous manuscript due to the insignificant effect. This is unprecise. Thus we have added the results of control experiments (fresh medium without fungi) in our manuscript now. We show the results of control experiments in Figure 2, 3 and 6 and in Table 3. Thank you very much!
Point 34: Please provide HPLC 
profiles/peaks of detected oxalic, gluconic, and citric acids.
Response 34: There are 72 HPLC profiles in all including A. niger, P. tricolor bioleaching as well as their triplicate samples. It is difficult to show these profiles considering the length of manuscript. But we have added the detailed information about HPLC tests in section 2.6 as follows.
“The concentration of organic acids were determined using high performance liquid chromatography (HPLC, Agilent 1200) with a variable wavelength detector (210 nm) and Dikma-C18 column, mobile phase of 5 mM H2SO4, and flow rate of 0.5 ml/min at ambient temperature.”
Thank you!
Point 35: A lot on the grammar and vocab need to be improved.
Response 35: We have thoroughly revised our manuscript. The English grammar and vocab were revised by several experts. We can use the English editing service if you still do not satisfy with our manuscript. Thank you very much!
Round 2
Reviewer 1 Report
I am satisfied with the authors answers.
Author Response
Point 1: I am satisfied with the authors answers.
Response 1: Thank you very much for your work!
Reviewer 2 Report
Thank you very much for all performed work on the manuscript, there are still few points to be addressed:
-All microorganisms should be written in italics, especially inside the figures 2 & 3 (legends next to color symbols)
-Line 1182: "were difficult by fungi activity." difficult to what? something is missing here...
-This is not a correct English: "Efficiently eroding or destroying the silicon compound in RM particles should be researched in the future for the purpose of increasing the total metal leaching rate, e.g., combine chemical leaching (inorganic acids) or bio-desilication (bacillus mucilaginosus) into fungal bioleaching process." Please rephrase it (efficient erosion and destruction, should be investigated, Bacillus with a capital letter, etc.)
-Naturally, you are not expected to provide 72 HPLC profiles of all triplicates of A. niger, P. tricolor bioleaching. But please do show exemplary profiles of detected acids as peaks on the chromatograms at the corresponding retention time. It will give just more values to your manuscript and will show the depth of your analysis.
-After you removed entirely EDS point data, you should provide something instead, otherwise how does the reader know that you detected "solid layer of silicon minerals"?? If you mention "silicon compounds" in your manuscript, you need to show how you determine that they are silicon compounds. Otherwise it is too semantic... If you performed several EDS measurements in that area, just do a statists of your measurements (mean, SD, n as number of measurements) and present a table with elemental analysis for raw and bioprocessed areas. Alternatively, go for XRD analysis, as you like.
Author Response
Point 1: All microorganisms should be written in italics, especially inside the figures 2 & 3 (legends next to color symbols) 

Response 1: We have revised figure 2 and figure 3. We have also thoroughly checked in our manuscript. Thank you!
Point 2: Line 1182: "were difficult by fungi activity." difficult to what? something is missing here...
Response 2: We have completed this sentence as “It indicated that the iron oxides were easy to be eroded while the silicon compounds were difficult to be eroded or destroyed by fungi activity”. Thank you!
Point 3: This is not a correct English: "Efficiently eroding or destroying the silicon compound in RM particles should be researched in the future for the purpose of increasing the total metal leaching rate, e.g., combine chemical leaching (inorganic acids) or bio-desilication (bacillus mucilaginosus) into fungal bioleaching process." Please rephrase it (efficient erosion and destruction, should be investigated, Bacillus with a capital letter, etc.)
Response 3: We have revised this sentence as “For the purpose of increasing the total metal leaching rate, the efficient erosion and destruction of silicon compounds in RM particles should be investigated in the future, e.g., combine chemical leaching (inorganic acids) or bio-desilication (Bacillus mucilaginosus) into fungal bioleaching process.” Thank you very much!
Point 4: Naturally, you are not expected to provide 72 HPLC profiles of all triplicates of A. niger, P. tricolor bioleaching. But please do show exemplary profiles of detected acids as peaks on the chromatograms at the corresponding retention time. It will give just more values to your manuscript and will show the depth of your analysis.
Response 4: We have added one exemplary chromatogram of HPLC in Figure 4 in our manuscript. We have also revised the first paragraph in section 3.4.3 as follows.
“The HPLC chromatogram of P. tricolor after bioleaching of 30 days is shown in Fig. 4. A large number of peaks appeared on the chromatogram, which indicated that various kinds of metabolites were secreted by fungi. The two highest peaks on the chromatogram represented oxalic and citric acids, respectively. The peaks of gluconic acids were inconspicuous due to the tiny production.”
Thank you very much for your suggestion!
Point 5: After you removed entirely EDS point data, you should provide something instead, otherwise how does the reader know that you detected "solid layer of silicon minerals"?? If you mention "silicon compounds" in your manuscript, you need to show how you determine that they are silicon compounds. Otherwise it is too semantic... If you performed several EDS measurements in that area, just do a statists of your measurements (mean, SD, n as number of measurements) and present a table with elemental analysis for raw and bioprocessed areas. Alternatively, go for XRD analysis, as you like.
Response 5: We have added Table 3 to show the EDS results. And we also revised this content in section 3.5. Thanks a lot for your correction!
